# Hyperoxidized Species of Heme Have a Potent Capacity to Induce Autoreactivity of Human IgG Antibodies

**DOI:** 10.3390/ijms24043416

**Published:** 2023-02-08

**Authors:** Marie Wiatr, Maya Hadzhieva, Maxime Lecerf, Rémi Noé, Sune Justesen, Sébastien Lacroix-Desmazes, Marie-Agnès Dragon-Durey, Jordan D. Dimitrov

**Affiliations:** 1Centre de Recherche des Cordeliers, INSERM, CNRS, Sorbonne Université, Université Paris Cité, 75006 Paris, France; 2Immunitrack Aps, Lersoe Park Alle 42, 2100 Copenhagen, Denmark; 3Service d’Immunologie Biologique, Hôpital Européen Georges Pompidou, Assistance Publique-Hôpitaux de Paris, 75610 Paris, France

**Keywords:** heme, oxidative modifications, antibodies, autoreactivity, polyreactivity

## Abstract

The interaction of some human antibodies with heme results in posttranslational acquisition of binding to various self- and pathogen-derived antigens. The previous studies on this phenomenon were performed with oxidized heme (Fe^3+^). In the present study, we elucidated the effect of other pathologically relevant species of heme, i.e., species that were formed after contact of heme with oxidizing agents such as hydrogen peroxide, situations in which heme’s iron could acquire higher oxidation states. Our data reveal that hyperoxidized species of heme have a superior capacity to heme (Fe^3+^) in triggering the autoreactivity of human IgG. Mechanistic studies demonstrated that oxidation status of iron was of critical importance for the heme’s effect on antibodies. We also demonstrated that hyperoxidized heme species interacted at higher affinities with IgG and that this binding occurred through a different mechanism as compared to heme (Fe^3+^). Regardless of their profound functional impact on the antigen-binding properties of antibodies, hyperoxidized species of heme did not affect Fc-mediated functions of IgG, such as binding to the neonatal Fc receptor. The obtained data contribute to a better understanding of the pathophysiological mechanism of hemolytic diseases and of the origin of elevated antibody autoreactivity in patients with some hemolytic disorders.

## 1. Introduction

Heme is a coordination complex of the macrocyclic compound protoporphyrin IX with iron. It is used as a prosthetic group by a large group of proteins referred to as hemoproteins. The iron ion in heme can easily accept and release electrons, thus alternating between ferrous (Fe^2+^) and ferric (Fe^3+^) oxidation state. This property along with the potential of heme to bind to some gaseous molecules explains its frequent use as a prosthetic group by proteins implicated in oxidative metabolism [1]. In addition to its firmly bound form, heme can also establish transient interactions with a considerable number of intracellular and extracellular proteins. In this manner, heme regulates diverse processes such as gene expression, cell signaling, cell cycle, and membrane transport of ions, etc. [2,3,4].

Under physiological conditions, heme is sequestered intracellularly. However, as a result of different pathological processes, large quantities of hemoproteins and heme can be released in the plasma [5,6]. The principal process leading to the release of heme is hemolysis [7]. Compelling evidence indicates that the heme liberated in the extracellular compartment manifests toxic effects. Thus, heme is prooxidative and acts as a danger signal, triggering inflammatory programs in different types of immune and nonimmune cells [8,9,10,11,12,13,14,15]. The proinflammatory potential of heme was proposed to contribute to the morbidity in a vast spectrum of diseases, such as sickle cell disease (SCD), malaria, sepsis, β-thalassemia, or intracranial hemorrhage [16,17,18,19,20]. In addition to affecting different cell types, there is evidence that free heme alters the functional activity of essential plasma proteins (reviewed in [21]). For example, heme can activate the complement system [22,23,24,25]. Importantly, heme can also influence antigen-binding functions of antibodies (Abs). Thus, the exposure of human serum or of purified polyclonal human IgG to heme results in the appearance of novel antigen-binding specificities [26,27,28,29,30]. It was documented that the contact with heme triggers binding reactivity of Abs to prototypical autoantigens such as nuclear proteins, phospholipids, β2-glycoprotein I, glutamic acid decarboxylase, myeloperoxidase, thyroglobulin, etc. [27,28,31]. Interestingly, immunoglobulin samples (that were) treated in vitro with heme satisfied the clinical diagnostic criteria for disease-associated autoantibodies [26,27,28]. Studies on monoclonal human IgGs revealed that in vitro interaction with heme can induce a single Ab molecule to bind multiple unrelated antigens, i.e., heme is able to confer polyreactivity [29,30,32,33]. It was shown that only a fraction of the Abs is sensitive to heme and that the changes in the antigen-binding specificity are the consequence of a direct binding of heme to the sensitive Abs [32,33]. Nevertheless, the molecular mechanism underlining heme-induced changes in antigen-binding specificities of Abs are poorly understood.

Available studies of heme effect on Abs were performed with the most stable oxidized form of heme, ferriprotoporphyrin IX. Notably, hemolysis can be accompanied by oxidative stress and production of reactive oxygen species, which exposes extracellular heme to further oxidative modifications [34,35]. Indeed, exposure of heme to neutrophil-derived hydrogen peroxide generates hyperoxidized forms of heme such as ferry and perferryl species [36]. Importantly, nonenzymatic heme degradation products with typical fluorescence properties have been reported in patients with SCD and in experimental models of hemolysis [36,37]. Despite their presence in pathological conditions, little is known about the way different oxidized heme species affect the specificity of human Abs. Studies of their effect may contribute to a better understanding the mechanism of heme-induced Ab polyreactivity and the pathophysiological relevance of the phenomenon. In the present study, we examined the effects of hyperoxidized species of heme on the reactivity of human IgG.

## 2. Results

### 2.1. The Interaction of Heme (Fe^3+^) with Prooxidative Compounds Increases Its Capacity to Induce the Autoreactivity of Human IgG

Previous studies on the effect of heme on the reactivity of human Abs have been performed by using the oxidized (Fe^3+^) form of heme. This form is relevant in hemolytic conditions due to the spontaneous autoxidation of reduced heme (Fe^2+^) in extracellular hemoglobin [6]. To elucidate whether other, in vivo-relevant, species of heme can impact the reactivity of human IgG, we pretreated oxidized heme (Fe^3+^) with increasing concentrations of hydrogen peroxide. Hydrogen peroxide is the most abounded oxidant in vivo and its release has been associated with hemolytic conditions [34,35]. As a result of their pseudoperoxidase catalytic activity, heme and some hemoproteins catalytically degrade hydrogen peroxide, a process that can be accompanied by formation of hyperoxidized heme species [36,38,39,40,41]. As previously described, [28,29] the exposure of human polyclonal IgG to heme (Fe^3+^) resulted in an increase of Ab binding to human proteins (Figure 1a). Next, we used heme that had been preincubated with increasing concentrations of hydrogen peroxide prior to exposure to human IgG. Heme pretreated with H_2_O_2_ had a significantly increased (*p* < 0.0001 one-way ANOVA, Kruskal–Wallis’s test) capacity to induce the autoreactivity of human polyclonal IgG as compared to native (Fe^3+^) heme (Figure 1a). The preexposure of heme to subequimolar concentrations of H_2_O_2_ (such as 0.24:1 H_2_O_2_:heme molar ratio) was sufficient to augment its ability to induce IgG autoreactivity. Of note, heme that was incubated with a large excess of hydrogen peroxide (>200-fold) presented with a slightly decreased, but still superior potency than intact heme (Fe^3+^), to induce the autoreactivity of polyclonal IgG (Figure 1a). The pretreatment of heme (Fe^3+^) with another in vivo-relevant oxidant, hypochlorite anions, also resulted in a significantly elevated potential to induce autoreactivity of Abs (Figure 1b, *p* < 0.01, one-way ANOVA, Kruskal–Wallis’s test). However, the incubation with larger molar excesses of hypochlorite resulted in a rapid loss in the capacity to induce IgG autoreactivity (Figure 1b). For our following experiments, we consistently used heme pretreated with a moderate molar excess of hydrogen peroxide (25:1 H_2_O_2_:heme ratio). This treatment is in the range of concentrations that substantially potentiated the capacity of heme (Fe^3+^) to induce IgG autoreactivity (Figure 1a). We refer to this heme species in the current work as “heme-ox”. Heme-ox demonstrated superior capacity than heme (Fe^3+^) to induce autoreactivity of polyclonal IgG toward human proteins in a large concentration range (Figure 1c, *p* < 0.001 two-way ANOVA test). ELISA and immunoblot analyses also showed that exposure of IgG to heme-ox generated significant reactivities to an extended panel of different plasma and intracellular proteins (Figure 1d,e). As compared to intact heme, heme-ox was characterized by a considerably higher potential to trigger the autoreactivity of human Abs (Figure 1d,e). Apart from the recognition of proteins, the treatment of IgG preparation with heme or with heme-ox resulted in a reduction of the intrinsic reactivity of human polyclonal IgG to nonproteinous antigens, i.e., DNA and LPS (Figure 1d). The indirect immunofluorescence analysis of reactivity of IgG to human neutrophils is a widely used diagnostic assay (ANCA assay) for the detection and characterization of pathogenic autoantibodies found in autoimmune diseases, mainly in systemic vasculitis. The exposure of polyclonal IgG to heme or to heme-ox resulted in an appearance of a strong reactivity to neutrophils (Figure 1f,g). The effect of heme-ox on IgG was higher as compared with that of intact heme (Fe^3+^). When studying the specificity of ANCA positivity, we observed a high reactivity against lactoferrin and not against other enzymes present in neutrophiles granules (MPO, PR3, elastase, cathepsine G and BPI). However, the antilactoferrin reactivity was similar between Abs that were exposed to heme (Fe^3+^) and to heme-ox (ratio of 3.9 and 3.7, respectively). Therefore, the increased reactivity against neutrophilis of heme-ox-treated IgG may be explained by a gain of binding specificities towards other autoantigens expressed by neutrophils.

Taken together these data indicate that molecular species of heme that have undergone oxidative reactions demonstrate a higher potential than heme (Fe^3+^) to posttranslationally uncover the autoreactivity of polyclonal human IgG.

### 2.2. Heme (Fe^3+^) and Heme-Ox Do Not Affect the Overall Molecular Integrity of IgG

We elucidated the impact of heme (Fe^3+^) and heme-ox on the molecular properties of polyclonal IgG. Size-exclusion chromatography ruled out the formation of high molecular weight species of IgG following treatment with heme (Fe^3+^) or heme-ox (Figure 2a). Importantly, when purified, the monomeric fraction of heme (Fe^3+^)/heme-ox-treated IgG retained their significantly increased potential for recognition of self-antigens (*p* < 0.005, one-way ANOVA, Kruskal–Wallis’s test; see Figure 2b). Next, we assessed the effect of heme (Fe^3+^) and heme-ox on the Fc-mediated functions of IgG. To this end, the kinetics of interaction of native, heme (Fe^3+^)- or heme-ox-treated Fcγ fragments (obtained by papain digestion of polyclonal human IgG) with immobilized human recombinant FcRn was evaluated by a surface plasmon resonance-based assay. Biosensor analyses demonstrated that heme (Fe^3+^) or heme-ox had no effect on the binding kinetics and affinity of interaction of Fcγ fragments with FcRn (Figure 2c).

Finally, by using protein carbonyls assay, we assessed whether exposure to heme-ox results in a major oxidative modification of the IgG molecule. Both heme (Fe^3+^) and heme-ox induced a marginal and insignificant increase (*p* = 0.25 Wilcoxon test) in the levels of carbonyls in IgG (Figure 2d). Conversely, addition of large excesses of hydrogen peroxide to a mixture of IgG and heme (Fe^3+^) considerably elevated the concentration of carbonyls (Figure 2d).

Collectively, these data suggest that the effect of heme (Fe^3+^) and heme-ox on the binding characteristics of human IgG is not the result of aggregation or other types of major alteration in the integrity of the immunoglobulin structure. Moreover, the data suggest that the effect of the heme species is constrained to the variable region of the Abs and does not affect the Fc fragment.

### 2.3. Consequences of the Exposure of Heme (Fe^3+^) to Hydrogen Peroxide

To understand the mechanism(s) underpinning the elevated potential of heme-ox to induce autoreactivity in polyclonal IgG, we next elucidated the molecular species generated upon interaction of H_2_O_2_ with heme. By applying UV-vis absorbance spectroscopy analysis, we demonstrated that the titration of heme with H_2_O_2_ resulted in a concentration-dependent reduction of the characteristic UV-vis spectrum of heme (Fe^3+^) in PBS (Figure 3a). The reduction of the absorbance of heme (Fe^3+^) was consistent with a nonenzymatic degradation of the macrocyclic ring of the porphyrin molecule. More specifically, these data showed that exposure of heme (Fe^3+^) to a high molar excess (1000×) of H_2_O_2_ resulted in a complete degradation of the porphyrin molecule. Importantly, the reaction of heme with a 25-fold molar excess of H_2_O_2_ resulted in only ~30% reduction in the absorbance at the Soret region and in only minimal changes in the low-energy region of the spectrum. Of note, the products of heme oxidation that are characterized by a considerable loss of absorbance in UV-vis absorption spectra had a suboptimal capacity to induce the autoreactivity of IgG Abs (Figure 1a). This result suggested that the molecular species of heme that most efficiently induced IgG autoreactivity retained to a large extent the integrity of the macrocyclic porphyrin structure.

Heme (ferriprotoporphyrin IX) has a pseudoperoxidase catalytic activity [38,39]. Indeed, we observed that the addition of a 25-fold molar excess of H_2_O_2_ to a heme stock solution (1 mM) was accompanied by an immediate evolution of gas, evidence for the catalytic degradation of peroxide to molecular oxygen and water. The catalytic degradation of H_2_O_2_ by heme has been shown to proceed by the formation of high-valent species of iron, i.e., Fe^IV^=O ferryl radicals [36]. The latter species possess a superior oxidizing capacity as compared to ferri (Fe^3+^) heme. Hence, one can speculate that the oxidation of heme by a moderate excess of peroxide results in the formation of ferryl heme species, which also have a superior capacity to modulate the reactivity of human IgG. To test this hypothesis, we first used absorbance spectroscopy. Cyanide anions (CN^−^) represent a high affinity ligand for heme (Fe^3+^). Indeed, addition of excess of CN^−^ resulted in marked changes in the UV-vis spectrum of heme, consistent with formation of a high-spin coordination complex (Figure 3b). Although heme-ox retained a substantial absorbance in the Soret region, the presence of an excess of CN^−^ did not result in significant spectral changes, as those manifested by the heme (Fe^3+^) (Figure 3b). This result implied that the iron ion of heme-ox had altered its oxidation state and coordination preferences and, as consequence, its capacity to interact with CN^−^ anions.

Furthermore, to rule out the possibility that traces of hydrogen peroxide, nondegraded by the pseudoperoxidase activity of heme, contributed to the modifications in the binding reactivity of IgG, we treated polyclonal IgG with a metalloporphyrin containing Zn(II), a redox inert metal. Zn(II) protoporphyrin IX was preincubated with a 25-fold molar excess of H_2_O_2_. Exposure of IgG to this porphyrin did not result in increased antigen binding reactivity (Appendix A). This result further highlights the essential role of the iron ion in heme for the induction of autoreactivity of IgG. Likewise, the absence of an effect of large concentrations of H_2_O_2_ on IgG autoreactivity rule out the possibility for a synergistic effect of heme and nondegraded H_2_O_2_ (Appendix A).

To elucidate the impact of the oxidation state of iron ion in heme-ox on the binding properties of Abs, we used ascorbic acid as a reducing agent. Ascorbate has a well-characterized potential to reduce hyperoxidized ferryl-heme radical to heme (Fe^3+^) [42]. Treatment with ascorbate resulted in a significant decrease in the IgG autoreactivity induced by heme-ox (*p* < 0.001 Two-way ANOVA test, Figure 3c). This result suggested that a higher oxidation potential of heme-ox had a direct contribution to the triggering of autoreactivity of IgG.

We conclude that the reaction of heme (Fe^3+^) with moderated excess of hydrogen peroxide affects the oxidation state of heme’s iron with only a minimal degradation of the macrocyclic ring. Changes in the oxidation state of the iron ion might play an essential role in the ability of heme molecule to modulate the reactivity of the human Abs.

### 2.4. Heme-Ox Interacts with IgG in a Different Manner as Compared to Heme (Fe^3+^)

To better understand the molecular aspects of interaction of heme-ox with IgG, we performed absorbance spectroscopy analyses. The binding of heme to polyclonal human IgG or to a prototypic monoclonal human IgG1 Ab21, which was previously shown to gain polyreactivity after contact with heme [30], was assessed by differential spectral analyses (Figure 4a). These analyses revealed that heme (Fe^3+^) established direct contacts with the protein, as evidenced by detection of an increased absorbance in the Soret region, a red shift to 414 nm (in the case of Ab21) of the Soret absorbance maximum, and spectral changes in the low energy region, i.e., the appearance of a characteristic peak around 640 nm (Figure 4a). These specific spectral changes indicated that heme bound to the IgG molecule and that this binding was accompanied by metal-coordinative interactions with certain amino acid residues, most probably His or Tyr. These results confirm our preceding studies [29,30]. As compared to heme (Fe^3+^), heme-ox generated considerably different spectral changes while binding to polyclonal or monoclonal human IgG (Figure 4a). Thus, the red shift in the Soret spectral region was less pronounced, especially in case of polyclonal IgG where a difference of 10 nm was observed (peak at 401 versus 411 nm). Moreover, the spectrum of heme-ox when bound to IgG differed profoundly in the low-energy region (500–700 nm) from those of heme (Fe^3+^) (Figure 4a). Thus, the binding of heme-ox to IgG occurred by a different mechanism as compared to that of heme (Fe^3+^). It can be deduced that heme-ox binds in a different molecular environment, with a distinct metal coordination pattern.

To better characterize the interaction of heme (Fe^3+^) and heme-ox with Abs, we compared the real-time binding of both species to immobilized monoclonal human IgG1 (Ab21) by surface plasmon resonance. The results (Figure 4b) demonstrated that heme-ox bound human IgG with a substantially higher affinity (23-fold increase), as compared with the affinity of heme (Fe^3+^). This result further substantiated the data from absorbance spectroscopy analyses, indicating that both heme species interacted with IgG by dissimilar mechanism.

In summary, data from this part suggest that the considerably increased capacity of heme-ox to induce the autoreactivity of IgG might be explained by an elevated binding affinity of heme-ox to IgG that takes place in a different molecular environment.

## 3. Discussion

In the present study, we showed that treatment of polyclonal human IgG with hyperoxidized heme species, referred here to as “heme-ox”, displayed a higher potential than heme (Fe^3+^) in inducing the autoreactivity of Abs. Heme-ox bound to IgG with an enhanced affinity as compared to heme (Fe^3+^) and used a distinct binding mechanism.

In vitro exposure of heme (Fe^3+^) to prooxidative substances such as hydrogen peroxide can result in minor oxidative modifications or in a complete degradation of the metalloporphyrin ring, depending on the particular reaction conditions [36,43,44,45,46,47,48]. The most abundant products of complete nonenzymatic degradation of heme are meleimides, propentdyopents, and hematinic acid [36]. These compounds are detected in vivo under different disease conditions and, similarly to extracellular heme (Fe^3+^), they have potent physiological activities and pathogenic potential [36,37,49,50,51,52]. Conversely, the contact of heme (Fe^3+^) with moderated excesses of hydrogen peroxide results in the formation of Fe^IV^=O or Fe^IV^-OH ferryl radicals that represent a hyperoxidized state of the iron ion. Such oxidation products with altered oxidation state of heme iron are not less relevant to pathology. Indeed, ferrylhemoglobin was shown to have substantially augmented proinflammatory activity in vitro and in vivo than methemoglobin (that contains heme Fe^3+^) [53].

In the present study, we used mostly species of heme generated after exposure to a moderate (i.e., 25-fold) molar excess of hydrogen peroxide. Our experimental data suggested that, at this concentration, the oxidant did not result in complete disintegration of the porphyrin structure. Indeed, an ultimate degradation of the heme molecule requires considerably harsher reaction conditions [44]. The moderate excess of hydrogen peroxide used in the present work would more likely favor the generation of hyperoxidized iron states. As a result of pseudoperoxidase catalytic reaction, it is also possible formation of heme species with covalent modifications of porphyrin ring due to radical attack [36,45]. An important finding from our study is that these species have a superior potential than heme (Fe^3+^) to induce a posttranslational increase in the autoreactivity of human Abs. Importantly, the impact of heme-ox was limited to the antigen-binding functions of IgG and interactions mediated by the Fc portion of the molecule were not modified by the treatment. The interaction of heme (Fe^3+^) with another physiologically relevant oxidizing agent, hypochlorite, also potentiated its capacity to modify the specificity of human Abs. However, the effect was rapidly lost at high excesses of OCl^−^ anions, most probably due to the ultimate degradation of heme moiety by such a potent oxidizing agent, as described before [54].

Previous work had demonstrated that the exposure of Abs to heme (Fe^3+^) induces the appearance of reactivities toward autoantigens and different pathogen-derived antigens [26,27,28,29,32]. Experiments with monoclonal Abs had shown that the contact of the Fab fragment with heme triggers polyreactivity [30,32,33]. This phenomenon was reported to occur in 10–20% of Abs in a normal immune repertoire and described for IgG, IgM, IgA, and IgE Abs [27,28,30,31,32,33,55,56,57]. Notably, certain heme-treated monoclonal Abs recognized their newly acquired target antigens with high binding affinity (K_D_~1–2 nM) [55]. The molecular mechanism underlying the appearance of novel antigen-binding reactivities of Abs induced by heme (Fe^3+^) is not well understood. It was established that induction of polyreactivity is a consequence of the binding of the heme molecule to the variable region of Ab [29,30,55]. The binding of heme (Fe^3+^) as well as the potential of Abs to acquire polyreactivity correlate with specific amino-acid sequence features of the antigen-binding site, such as the presence of significantly higher numbers of Tyr, Lys, or Arg residues, and significantly lower numbers of Glu and Asp residues in the CDR loops [33]. These sequence traits favor a molecular microenvironment that is pertinent for the binding of heme molecule, which exhibits high hydrophobicity and contains two negatively charged carboxyl groups. Furthermore, prior studies established that the presence of the iron ion in the macrocyclic structure of heme is indispensable for its effect on Abs [30]. Molecular modeling and binding assays suggested that heme while bound to Ab is exposed on the molecular surface [33,55]. Hence, it was speculated that heme functions as an interfacial bridge between the Ab molecule and newly recognized albeit noncognate antigens [58]. This hypothesis is supported by the recent studies showing that free heme demonstrates a high intrinsic binding promiscuity towards human and bacterial proteins [59,60,61].

In addition to its role as an interfacial cofactor for antigen binding, the prooxidative potential of heme (Fe^3+^) may also be implicated in the gain of novel antigen reactivities by Abs. This stems from findings that the exposure of Abs to other prooxidative agents (including iron ions or various ROS) also triggers autoreactivity and polyreactivity, albeit at a lower extend than heme (Fe^3+^) [28,62,63]. Consistently, as observed here, the heme species with a higher oxidative potential, i.e., ferryl-heme, had a superior capacity to induce Abs autoreactivity. However, our carbonyl assay did not demonstrate major oxidation of IgG by heme-ox, suggesting that the oxidative modifications, if they occur, are limited to the antigen-binding site and are minor.

The significance of the oxidation status of iron ion in heme-ox for its effect on Abs was directly proven by the reversion of the higher oxidation state through incubation with a reducing agent (ascorbic acid). Ascorbic acid reverts the ferryl state of heme’s iron to Fe^3+^ form [42]. Indeed, the incubation of heme-ox with ascorbate resulted in a reduction of its capacity to induce autoreactivity of human Abs. Nevertheless, heme-ox was found to bind with a significantly higher affinity (23×) to IgG molecules as compared to heme (Fe^3+^). A higher affinity of the heme-Ab complex is in favor of a more efficient use of heme moiety as an interfacial cofactor in antigen recognition. It is noteworthy, that ferryl-species of heme are unstable. Thus, in absence of residual peroxide they might not be available for an extended time in the reaction system. Therefore, one should not exclude the possibility that species of heme with covalent modifications of the porphyrin ring, which might exhibit better binding affinity for antibodies, could also be responsible for the observed effects. It is also possible that exposure of heme to moderate excesses of hydrogen peroxide results in heterogenous mixture of heme species, which as a whole have a higher potential to induce autoreactivity of antibodies. Further studies should clarify the exact chemical nature of heme species generated in our experimental condition as well as whether the increased binding to IgG molecule, the elevated oxidative potential or both together contribute to the potent effect of hyperoxidized heme species on human IgG.

In parallel, as its effect on antibodies, a previous study showed that heme-ox exerts higher biological activity in a more complex system. Thus, an exposure of human endothelial cells to heme-ox resulted in potent activation of the complement system on the cellular surface [64]. It is important to understand whether similar modifications in heme that cause more potent effects on antibodies are implicated in the effect of heme-ox in cellular system. In this study was also demonstrated that human therapeutic pooled immunoglobulin G (IVIG) could directly scavenge heme-ox and protect endothelial cells from complement attack [64].

Pathological situations associated by the release of large quantities of extracellular heme are often accompanied by oxidative stress [34,35,65,66,67,68]. Under inflammatory conditions, hydrogen peroxide and hypochlorous acid are generated in large quantities by polymorphonuclear cells [69]. Accordingly, it has been observed that in hemolytic conditions, oxidized forms of heme are generated in vivo [36,37,41,51]. Therefore, it is most likely that in such pathological situation Abs experience the effects not only of heme (Fe^3+^) but also of its oxidation derivatives. It is noteworthy that diseases associated with hemolysis, such as SCD and malaria, are often accompanied by the increased prevalence of autoantibodies [70,71,72,73,74,75,76,77]. Moreover, patients with hemolytic diseases frequently present with kidney deposits of IgG and IgM Abs [78,79,80]. The origin of Abs directed to self-antigens and their contribution to the pathophysiology of hemolytic diseases with nonimmune etiology remain enigmatic. A plausible hypothesis is that part of the observed Ab autoreactivity in patients with hemolytic conditions reflects the functional impact of the interaction of heme and oxidized heme species with circulating Abs or with B-cell receptors. A posttranslational induction of autoreactivity of circulating Abs would result in the prompt formation of immune complexes with plasma proteins and deposition of these complexes in the glomeruli. In this regard, heme and its derivatives might express a pathogenic potential by evoking autoantibody reactivity that may contribute to the organ damage. Conversely, one should not exclude the possibility that the presence of Abs in healthy immune repertoires that acquire autoreactivity under exposure to heme and/or its oxidized metabolites might have protective effects by facilitating clearance of cell debris and macromolecules released in large quantities in situations of intravascular hemolysis.

In conclusion, hemolytic conditions witness the release of a plethora of hemolytic products—hemoglobin, heme (Fe^3+^), oxidative modified species of heme, as well as ROS. We demonstrated before that hemoglobin, an upstream product of hemolysis in relation to heme (Fe^3+^), has the capacity to trigger the autoreactivity and polyreactivity of human IgG through interprotein transfer of its heme moiety (article in press doi: 10.1038/s42003-023-04535-5). In the present study, we further demonstrate that oxidative products downstream from heme (Fe^3+^) have a potent capacity to modify the binding properties of human Abs. Thus, the present work completes the integrative exploration of the functional impact of various hemolysis byproducts on IgG reactivity. The results from this study may contribute to a better understanding of the pathophysiological consequences of hemolysis. Moreover, they provide important information that would contribute for deciphering the mechanism of heme-induced posttranslational acquisition of autoreactivity of Abs. Future studies with appropriate cellular and in vivo models should be performed to unravel the pathophysiological significance of the described phenomenon.

## 4. Materials and Methods

### 4.1. Materials

As a source of human polyclonal IgG obtained from healthy individuals, we used pooled immunoglobulin G preparation for therapeutic intravenous use (IVIG, Endobulin, Baxter International, Deerfield, IL, USA). We also used a previously described human recombinant monoclonal IgG1 (Ab21) [30].

A standard solution of hyperoxidized heme (heme-ox) was prepared by treatment of 1 mM of heme (Fe^3+^) or hemin (ferriproptoporphyrin IX, Frontier Scientific Inc., Logan, UT, USA) dissolved in 0.05 N NaOH with 25 mM final concentration of H_2_O_2_ (Merck, Darmstadt, Germany) in 0.05 N NaOH. Formation of gas bubbles after addition of H_2_O_2_ signified the catalytic degradation of peroxide to H_2_O and O_2_. For generation of oxidized species of heme, the samples were incubated in the presence of hydrogen peroxide for at least 10 min before any further use. In another setting, 1 mM hemin solution in NaOH was exposed to increasing concentrations (0.015–1000 mM, dilution step of 3) of hydrogen peroxide or sodium hypochlorite (Sigma-Aldrich, St. Louis, MO, USA). As a control, 1 mM solution of Zn(II) protoporphyrin IX (Frontier Scientific Inc.) in 0.05N NaOH was treated with 25 mM H_2_O_2_. In another case, heme (Fe^3+^) and heme-ox at 1 mM were reduced by incubation in the absence or in the presence of 10 mM ascorbic acid (Sigma-Aldrich) for 30 min at RT and then used for treatment of polyclonal IgG.

### 4.2. ELISA

All ELISA experiments were performed by using 96-well polystyrene plates (NUNC Maxisorp, Roskilde, Denmark). The plates were coated with human factor H (CompTech, Tyler, TX, USA), human factor IX (LFB, Les Ulis, France), human apo-hemoglobin, human transferrin, human collagen, bovine histone type III, equine cytochrome C, LPS from *E. coli* (O55:B5) and bovine double-strand DNA (all from Sigma-Aldrich) at final concentration of 10 μg/mL in PBS pH 7.4. Apo-hemoglobin was prepared from human hemoglobin by acetone extraction procedure. After incubation for 2 h at room temperature (RT), plates were blocked by incubation with 0.25% solution of Tween 20 in PBS. Polyclonal IgG (dialyzed against PBS) was diluted to 10 μM (1.5 mg/mL) in PBS and then incubated for 30 min on ice with 20 μM final concentrations of intact heme (Fe^3+^) or heme-ox. Following this incubation step, IgG samples were serially diluted from 5 to 0.00195 μM in PBS, containing 0.05% Tween 20 (PBS-T) and incubated with antigen-coated plates for 2 h at RT. In another experimental setting, the IgG preparation diluted to 6.7 μM (1 mg/mL) in PBS was exposed to increasing concentrations of heme (Fe^3+^) or heme-ox (0, 0.125–32 μM) for 30 min on ice. Native and heme (heme-ox)-exposed IgG were further diluted to 50 μg/mL in PBS-T and incubated with antigen-coated plates for 2 h at RT. The effect of oxidation of 1 mM heme (Fe^3+^) by varying concentrations (0.015–1000 mM) of H_2_O_2_ or NaOCl was evaluated after incubation for 30 min on ice of 10 μM IgG with 20 μM of preliminary oxidized heme species. After this incubation, the samples were diluted to final IgG concentration of 1 μM (150 μg/mL) in PBS-T and incubated with immobilized antigens for 2 h at RT.

After incubation with immunoglobulins plates were washed with PBS-T and then further incubated for 1 h at RT with HRP-conjugated mouse anti-human IgG (clone JDC-10, Birmingham, AL, USA), diluted 3000 folds in PBS-T. Immunoreactivity of IgG was detected after addition of peroxidase substrate solution-o-phenylenediamine dihydrochloride (Sigma-Aldrich) and stopping the reaction by 2N HCl. The optical density at 492 nm was read by using TECAN Infinite M200Pro microplate reader.

### 4.3. Immunoblot

Lysate from human umbilical vain endothelial cells (in reducing sample buffer) was loaded on gradient (4–12%) Bis-tris gels (Life technologies, Novex NuPAGE gels, Thermo Fisher Scientific, Waltham, MA, USA) with a total protein amount of 100 μg/gel and proteins were separated in ready-made MES-SDS running buffer (Life technologies, NOVEX, Thermo Fisher Scientific) at constant voltage of 200 V. After electrophoretic separation, proteins were transferred on nitrocellulose membranes (iBlot gel transfer stacks, Invitrogen, Thermo Fisher Scientific) by using iBlot electrotransfer system (Invitrogen, Thermo Fisher Scientific). Nitrocellulose membranes were blocked overnight in TBS, containing 0.1% Tween 20 (TBS-T) at 4 °C. Human IgG at 10 μM (1.5 mg/mL) was exposed to 20 μM of heme (Fe^3+^) or heme-ox in PBS and incubated for 30 min on ice. Next, the membranes were mounted on miniblot system (Immunetics, Cambridge, MA) and incubated with native, heme (Fe^3+^), and heme-ox treated IgG, diluted to 0.0312, 0.0625, 0.125, 0.25, 0.5, and 1 μM in TBS-T. Following an incubation for 2 h at RT, membranes were washed for 1 h with TBS-T (with at least six changes of the buffer). To detect protein-bound Abs, the membranes were probed for 1 h at RT with goat antihuman IgG, conjugated with alkaline phosphatase (Southern Biotech) diluted 3000× in TBS-T. After extensive washing with TBS-T for 1 h, immunoreactivity of IgG was revealed by addition of substrate solution SIGMAFAST BCIP/NBT (Sigma-Aldrich). Reaction was stopped by thoroughly washing membranes with deionized H_2_O.

### 4.4. Indirect Immunofluorescence ANCA Assay

(1)With regard to the preparation of samples, polyclonal IgG (PBS dialyzed) was first diluted to 10 mg/mL (67 μM) in PBS, and treated with 40 μM of final concentration of heme (Fe^3+^), heme-ox, or excipient only. After treatment, the preparations were stored at 4 °C until use.(2)ANCA assay was performed by indirect immunofluorescence by using biochip of human neutrophils fixed in ethanol and formol (Euroimmun, Lübeck, Germany) according to the manufacturer’s recommendations with samples diluted from 1/20 to 1/1280 in sample buffer. Biochip reading was performed with a fluorescent microscope (Axioscope A1, Zeiss, Oberkochen, Germany) and the positivity was assessed by two independent readers in a blind manner. Identification of the antigenic target was performed by ELISA (Euroimmun) by using ANCA profile assay (proteinase 3, myeloperoxydase, elastase, cathepsin G, bacterial permeability increasing protein, lactoferrin separately) according to the manufacturer‘s procedure. Results were expressed as the ratio between the optical density (OD) of the well of each antigen to the OD of the reference well, and positivity was defined by a ratio superior to 1.

### 4.5. Size Exclusion Chromatography

Molecular composition of native and heme/heme-ox exposed polyclonal human IgG was analyzed by using FPLC Akta Purifier (Cytiva, Marlborough, MA, USA), equipped with Superose 6 10/300 column. Human IgG preparation was diluted to 10 µM in PBS and exposed to 40 µM of heme or heme-ox. The samples were incubated for 15 min on ice. One ml of each native, heme (Fe^3+^) and heme-ox exposed IgG was loaded on column preequilibrated with PBS. The flow rate of 0.5 mL/min was used. Chromatograms were recorded by using UV detection of protein at wavelength of 280 nm. The monomeric IgG fractions were collected and their immunoreactivity analyzed by ELISA.

### 4.6. Determination of Oxidative Modifications in IgG (Carbonyls Assay)

Oxidative modifications of IgG molecules after the exposure to heme or heme-ox were assessed by dinitroohenylhydrazine colorimetric assay. Human IgG at 10 µM in PBS was treated with 10 µM heme (Fe^3+^), 10 µM heme-ox, and 1 mM H_2_O_2_ for 30 min at RT. IgG incubated in PBS only was used as a negative control. In alternative setting, IgG preparation that was first exposed to 10 µM heme (Fe^3+^) and then to 1 mM H_2_O_2_ was used as a positive control. After this, an equal volume of 10 mM solution of 2,4-dinitrophenylhidrazine (Sigma-Aldrich) in 2M HCl was added to each sample, and the samples were incubated for 1 h at RT with agitation on each 10 min. Trichloroacetic acid (Sigma-Aldrich), at a final concentration of 20%, was added to each sample to precipitate the protein. After centrifugation and threefold washing of the pellet with 1:1 solution of ethanol and ethyl acetate (Sigma-Aldrich), pellets were solubilized by addition of warm (37 °C) 6M solution of guanidine hydrochloride (Sigma-Aldrich) and incubated at 37 °C for 15 min. To quantify the protein carbonyls the optical density at 366 nm was estimated in 1-cm cuvettes. Estimation of concentration of carbonyls in the samples was performed by using molar extinction coefficient of 22,000 M^−1^ cm^−1^ for 2,4-dinitrophenyl at 366 nm.

### 4.7. UV-Vis Absorbance Spectroscopy

UV-visible spectra of heme were measured by UNICAM Helios β spectrophotometer. All measurements were performed in 1-cm quartz optical cells at RT. The absorbance spectra of intact heme (Fe^3+^) or heme exposed to increasing concentrations of H_2_O_2_ (0.015, 0.06, 0.24, 0.97, 3.91, 15.625, 62.5, 250, and 1000 mM) were recorded at the wavelength range of 350–700 nm, with a scan speed of 300 nm/min. The treatment of heme (Fe^3+^) with H_2_O_2_ was done after diluting hemin stock to 1 mM final concentration in 0.05N NaOH. For determination of absorbance spectra, heme was further diluted in PBS to 10 μM. In another experimental setting, heme (Fe^3+^) or heme-ox were diluted in PBS to 10 μM and treated for 5 min with 1 mM KCN (Sigma-Aldrich). Absorbance spectra of heme (Fe^3+^)/heme-ox in the absence or in the presence of KCN were recorded as described above.

For evaluation of heme (Fe^3+^)/heme-ox binding to IgG, human polyclonal- or monoclonal recombinant IgG1 (Ab21) were first diluted in PBS to 10 and 1 μM, respectively. Immunoglobulin solutions were titrated with increasing concentrations of heme or heme-ox–0.25, 0.5, 1, 2, 4 and 8 μM (in case of Ab21) and −0.5, 1, 2, 4, 8, and 16 μM (in the case of polyclonal IgG). Same aliquots of heme (Fe^3+^)/heme-ox were added to an optical cell, containing PBS only. After incubation for 2 min at RT spectra in the wavelength range, 350–700 nm were recorded. Evaluation of heme (Fe^3+^)/heme-ox binding to human IgG was performed after subtraction of spectra obtained of heme (Fe^3+^)/heme-ox in PBS from the spectra obtained in the presence of polyclonal IgG or monoclonal IgG1. For the sake of clarity, only the differential spectra obtained after addition of 16- and 8-μM, heme species to polyclonal IgG and Ab21, respectively, were displayed on Figure 4a.

### 4.8. Real-Time Interaction Analyses

#### 4.8.1. Measurement of Binding Affinity of Fc-γ Fragments to Human FcRn

For assessment of the effect of heme (Fe^3+^)/heme-ox on Fc-mediated binding to FcRn, we used an approach recently described in Ref. [81]. Recombinant human FcRn conjugated with biotin (Immunitrack; Copenhagen, Denmark, [82,83]) was immobilized on a strapatavidin sensor chip (SA chip, Biacore, Cytiva, Uppsala, Sweden) at a density of 1500 RU. Binding kinetics of Fc-γ fragments obtained by papain digestion of plyclonal human IgG was investigated by optical biosensor system—Biacore 2000 (Biacore, Cytiva). Native Fc-γ or Fc-γ treated in PBS at 12 µM with equimolar concentration of heme (Fe^3+^)/heme-ox were diluted serially (each step twofold) from 50 to 0.39 nM in 100 mM Tris-Citrate buffer pH 5.4, containing 0.1% Tween 20 and allowed to interact with immobilized FcRn for 4 min. Dissociation of the complex was followed for 5 min. The running buffer—100 mM Tris-Citrate buffer pH 5.4, containing 0.1% Tween 20, was filtered through 0.22 µm filter and degassed under vacuum. The sensor chip surfaces were regenerated by exposure to 100 mM Tris-Citrate buffer pH 7.4, containing 0.1% Tween 20 for 60 s. All kinetic measurements were performed at temperature of 25 °C. The evaluation of the kinetic data was performed by BIAevaluation version 4.1.1 Software (Biacore, Cytiva).

#### 4.8.2. Measurement of Binding Affinity of Heme to Human IgG

Kinetic analyses of interaction of heme (Fe^3+^)/heme-ox with human IgG was performed by surface plasmon resonance-based optical biosensor system—Biacore 2000 (Biacore, Cytiva). Human recombinant IgG1 (Ab21) was covalently immobilized on CM5 sensor chips (Biacore, Cytiva) by using an amine-coupling kit (Biacore). Briefly, IgG was diluted in 5 mM maleic acid (pH 3.85) to final concentrations of 10 μg/mL and injected over sensor surfaces activated by a mixture of 1-Ethyl-3-(3-dimethylaminopropyl)-carbodiimide (EDC) and N-hydroxysuccinimide (NHS). Uncoupled activated carboxyl groups were blocked by injection of 1M solution of enthanolamine.HCl. The running buffer PBS was filtered through 0.22 µm filter and degassed under vacuum. To evaluate the binding kinetics of the interactions of heme (Fe^3+^) and heme-ox with monoclonal IgG1, were serially diluted (each step twofold) in PBS to concentrations ranging from 1000 to 15.6 nM (heme-ox) and 5000 to 78.1 (heme Fe^3+^) and injected over immobilized IgG. The flow rate during all interaction analyses was set at 30 µL/min. The association and dissociation phases of the binding heme species were both monitored for 5 min. The sensor chip surfaces were regenerated by exposure to a solution of 300 mM imidazole for 30–60 s. All kinetic measurements were performed at a temperature of 25 °C. The evaluation of the kinetic data was performed by BIAevaluation version 4.1.1 Software (Biacore, Cytiva).

### 4.9. Statistical Analyses

The statistical analyses in the manuscript were performed by Graph Pad Prism 9.5 software (San Diego, CA, USA). The statistical significance of the difference between different experimental conditions was assessed by nonparametric one-way ANOVA, Kruskal–Wallis’s test, or two-way ANOVA test depending on whether one or more groups of data were compared. The paired samples were also tested by Wilcoxon test.

## Figures and Tables

**Figure 1 ijms-24-03416-f001:**
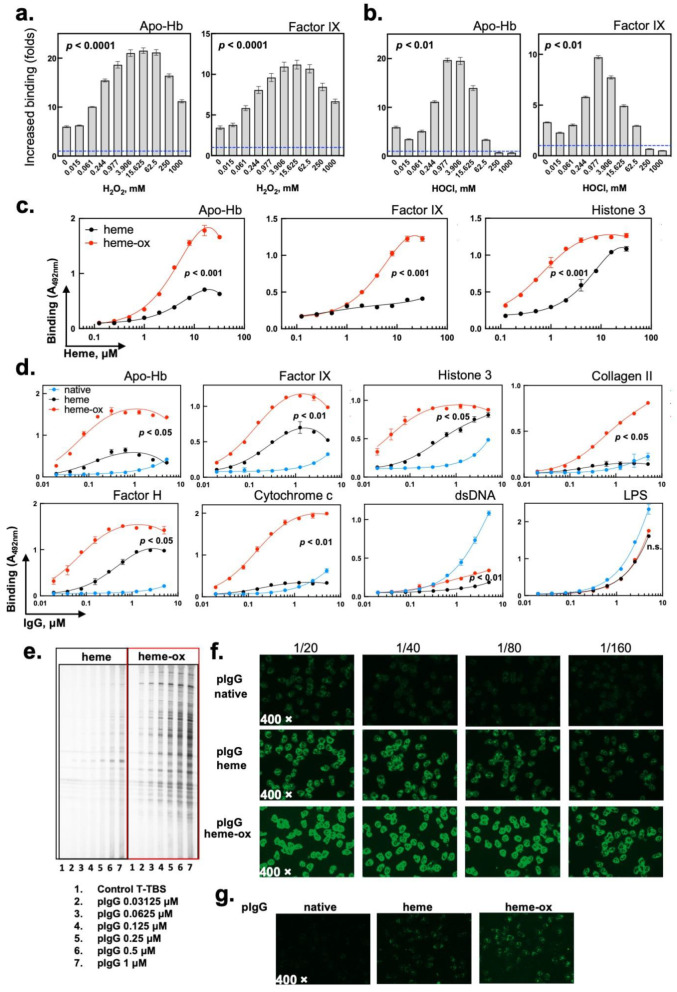
Heme broadens repertoire of recognized antigens by human IgG. Effect of oxidation of heme (Fe^3+^) by hydrogen peroxide (**a**) or of hypochlorite (**b**) on its ability to induce reactivity of human IgG to human apo-hemoglobin and Factor IX. Hematin at 1 mM in 0.05 N NaOH was first exposed to increasing concentrations of oxidants (0, 0.015–1000 mM). After polyclonal IgG at 10 μM was treated with 20 μM of intact or preoxidized heme and further diluted in PBS-T for incubation with immobilized antigens. The dashed blue lines on the graphs represents binding of native IgG the autoantigens. Each bar on the graphs represents mean optical density values ± SD (*n* = 6, for H_2_O_2_ and *n* = 3 for OCl^−^). The statistical significance of the difference between different conditions was assessed by one-way ANOVA and Kruskal–Wallis’s test. A representative result out of two independent experiments is shown. (**c**) ELISA analyses of the immunoreactivity of human IgG exposed to varying concentration of heme (Fe^3+^) or to heme-ox. ELISA analyses of immunoreactivity of polyclonal IgG towards apo-hemoglobin, factor IX, and histone 3. IgG at 6.7 μM (1 mg/mL) was incubated with increasing concentrations of heme/heme-ox (0, 0.125–32 μM). Each dot represents mean optical density values ±SD (*n* = 2) from technical replicates. The statistical significance of the difference between IgG treated with varying concentrations of heme (Fe^3+^) and heme-ox was assessed by two-way ANOVA, and Kruskal–Wallis’s test. (**d**) ELISA analyses of concentration dependence of binding of native-, heme (Fe^3+^)- and heme-ox-treated Abs to the indicated antigens. IgG at concentration of 10 μM (1.5 mg/mL) in PBS was exposed to 20 μM heme or to identical concentration of heme-ox and then serially diluted from 5 to 0.00195 μM in PBS, containing 0.05% Tween 20 (PBS-T). Each dot represents mean optical density values ± SD from (*n* = 3) technical replicates. The statistical significance of the difference between different concentration heme (Fe^3+^) and heme-ox-treated IgG was assessed by two-way ANOVA, and Kruskal–Wallis’s test. (**e**) Immunoblot analyses of reactivity of heme (Fe^3+^) and heme-ox treated polyclonal IgG. Human IgG at 10 μM (1.5 mg/mL) in PBS was exposed to 20 μM of heme (Fe^3+^) or heme-ox. Immunoglobulins were further diluted (0.0312, 0.0625, 0.125, 0.25, 0.5, and 1 μM) in TBS, containing 0.1% Tween 20 (TBS-T) and the binding to lysate of human endothelial cells was detected. A representative result from two independent experiments is shown. (**f**,**g**) ANCA assay—reactivity of native, heme (Fe^3+^) and heme-ox treated IgG to (**f**) ethanol or (**g**) formol fixed human neutrophils. All immunofluorescence images were acquired with 400× magnification.

**Figure 2 ijms-24-03416-f002:**
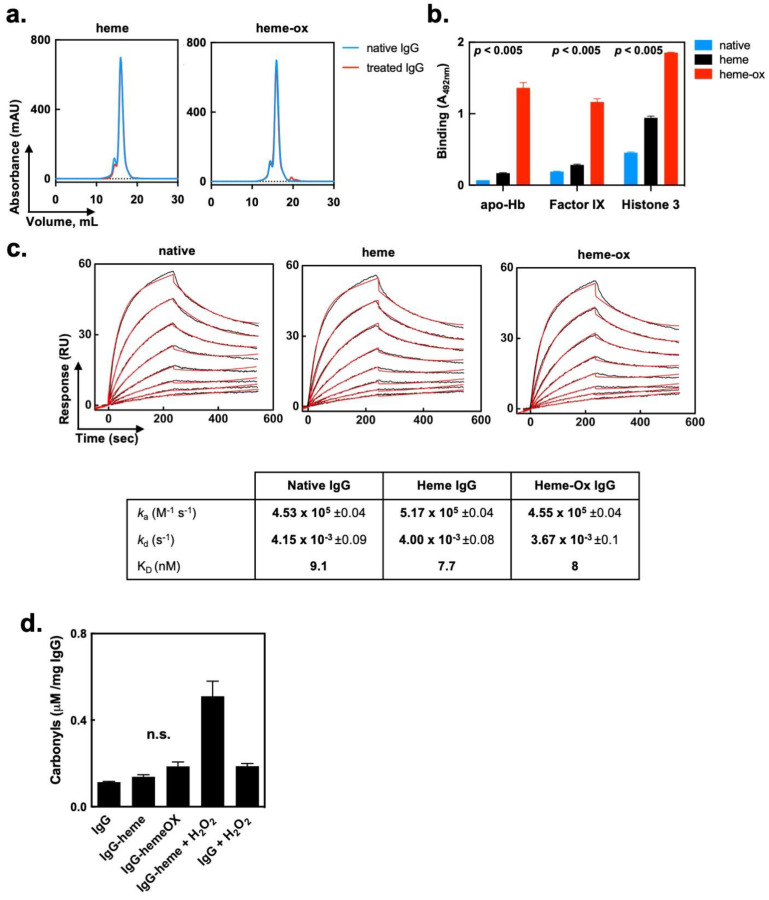
Impact of heme (Fe^3+^) and heme-ox on the molecular integrity and Fc-mediated functions of human IgG molecule. (**a**) Size-exclusion chromatography. Elution profiles of 10 μM of native polyclonal human IgG, and of 10 μM IgG preexposed to 40 μM of heme (left panel) or 40 μM heme-ox (right panel). (**b**) Enhanced reactivity of heme-ox treated IgG is not due to immunoglobulin aggregation. Immunoreactivity of the monomeric IgG fractions purified by size-exclusion chromatography from native polyclonal IgG, heme (Fe^3+^)-exposed IgG and heme-ox exposed IgG. The binding of native or treated IgG to immobilized human factor IX and human apo-hemoglobin was assessed by ELISA. Each bar represents mean optical density values ±SD (*n* = 3) from technical replicates. The statistical significance between the groups in cases of each studied antigen was assessed by one-way ANOVA and Kruskal–Wallis’s test. (**c**) Heme and heme-ox do not influence interaction of IgG with FcRn. Real-time interaction profiles of binding of native- (left panel), heme (Fe^3+^)- (central panel), and heme-ox-exposed Fc fragments from IgG. Fc fragments at 12 μM in PBS were exposed to 12 μM heme (Fe^3+^) or heme-ox. Furthermore, native and treated Fc were diluted in Tris-citrate buffer (pH 5.4) and injected over immobilized human recombinant FcRn at concentrations in the 50–0.39 nM. The black lines represent real time binding curves; the red lines represent the results of the fit generated by global analyses of experimental data. The kinetic analyses were performed at 25 °C. The kinetic parameters presented in the table were evaluated by global analyses by using BIAevaluation software. (**d**) Determination of oxidative modification in polyclonal IgG following exposure to heme (Fe^3+^)/heme-ox. Levels of carbonyls in (1) native IgG at 10 μM; (2) 10 μM of IgG treated with 20 μM heme (Fe^3+^); (3) 10 μM of IgG treated with 20 μM heme-ox; (4) 10 μM of IgG-treated with 20 μM heme followed by exposure to 1 M H_2_O_2_ and (5) 10 μM IgG treated with 1 M H_2_O_2_. Carbonyls were determined by derivatization of proteins with acidified dinitrophenylhydrazine and assessment of the optical density at 366 nm. Bars represent mean optical density values ±SD from (*n* = 3) technical replicates. The levels of carbonyls in heme (Fe^3+^) and heme-ox were not significant, as indicated by Wilcoxon test.

**Figure 3 ijms-24-03416-f003:**
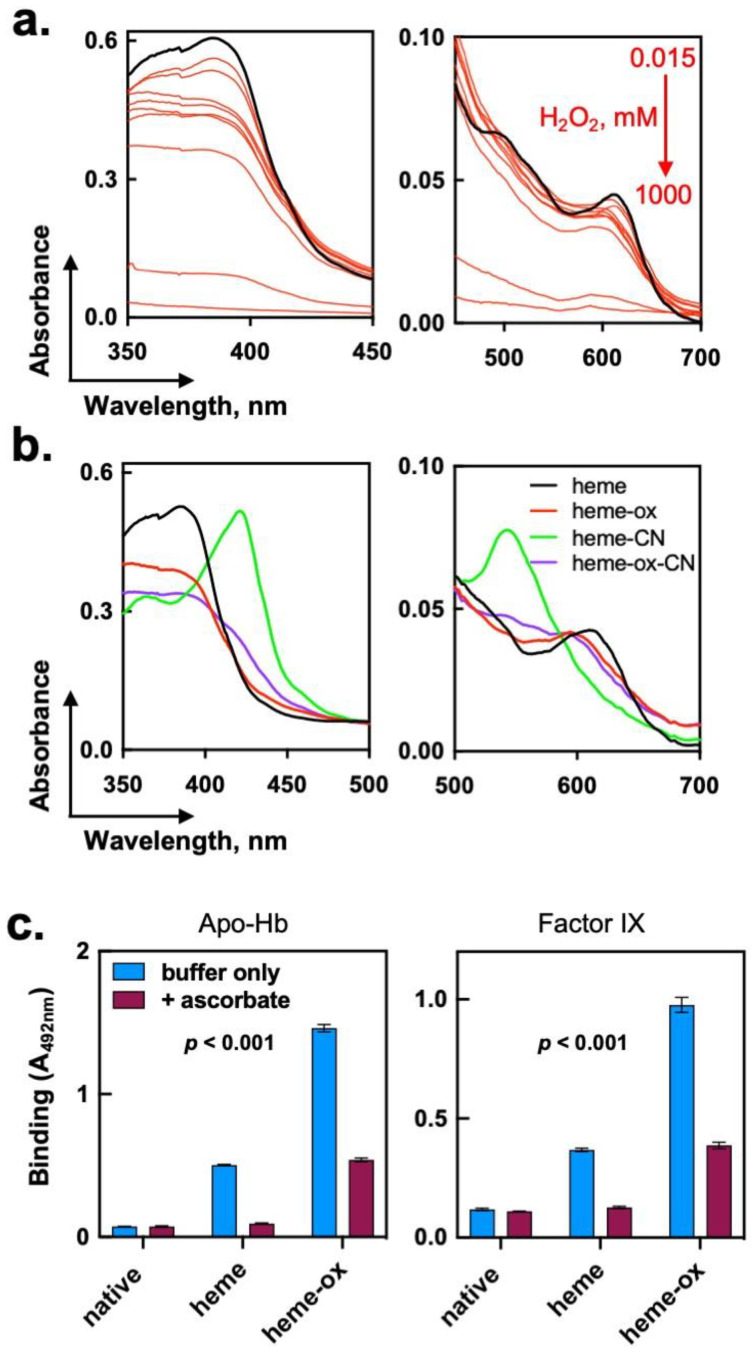
Assessment of structural and functional changes in heme (Fe^3+^) upon its oxidation with hydrogen peroxide. (**a**) Spectroscopic evaluation of oxidative modification of heme by H_2_O_2_. Heme at concentration of 1mM was first preexposed to increasing concentrations i.e., 0.015, 0.06, 0.24, 0.98, 3.91, 15.63, 62.5, 250, and 1000 mM of H_2_O_2_. Absorbance spectra of native (black line) and oxidized heme by different concentrations of H_2_O_2_ (red lines) were measured after dilution to 10 μM in PBS. The absorbance spectra were recorded in the wavelength range 350–700 nm in a quartz cuvette with 1 cm optical path. The data shown is a representative result out of two independent experiments. (**b**) Oxidation of heme (Fe^3+^) by H_2_O_2_ affects its coordination interactions. Absorbance spectra of heme (Fe^3+^) and heme-ox (both at concentration of 10 μM) in the absence or in the presence of potassium cyanide (1 mM). (**c**) ELISA analyses of potential of heme (Fe^3+^) and heme-ox to induce novel specificity of polyclonal IgG after reduction with ascorbic acid. Heme (Fe^3+^) was first oxidized by exposure to 25 molar excess of H_2_O_2_. Then, intact heme (Fe^3+^) and heme-ox at 1 mM were incubated in the absence or in the presence of 10 mM ascorbic acid. IgG was diluted to 10 μM and exposed to heme (Fe^3+^)/heme-ox ± ascorbic acid. After incubation for 30 min on ice, reactivity of Abs (150 μg/mL in PBS-T) toward human FIX and human apo-hemoglobin was assessed. Each bar on the graphs represents mean optical density values ± SD (*n* = 3) from technical replicates. The statistical significance of the difference between the groups of samples without and with ascorbic acid was assessed by two-way ANOVA, test.

**Figure 4 ijms-24-03416-f004:**
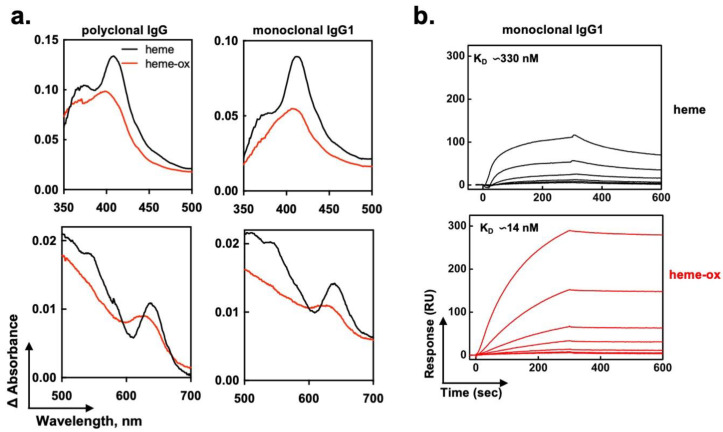
Mechanism of interaction of heme (Fe^3+^) and heme-ox with human IgG. (**a**) Differential spectra obtained after substruction of spectra of heme (Fe^3+^) and heme-ox in PBS from the respective spectra in the presence of human polyclonal IgG (left panels) or monoclonal IgG1 (Ab21, right panels). Polyclonal IgG or Ab21 were used at concentrations of 10 and 1 μM, respectively. Heme species were used at concentrations of 16 and 8 μM for treatment of polyclonal and monoclonal IgG, respectively. (**b**) Heme-ox binds with higher affinity than heme (Fe^3+^) to human IgG. Real-time SPR analyses of interaction of increasing concentrations of heme (Fe^3+^) (78.1–5000 nM) and of heme-ox (15.6–1000 nM) to immobilized human monoclonal IgG1 (Ab21). The kinetic analyses were performed at 25 °C. The kinetic parameters were evaluated by global analyses by using BIAevaluation software.

## Data Availability

The data that support the findings of this study are available from the corresponding author upon reasonable request.

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
