# Peer review of "Hyperoxidized Species of Heme Have a Potent Capacity to Induce Autoreactivity of Human IgG Antibodies"

_ijms, 2023, doi:10.3390/ijms24043416_

Round 1
Reviewer 1 Report
Review comments on ijms-2164251
Journal: International Journal of Molecular Sciences
Manuscript ID: ijms-2164251
Type of manuscript: Article
Title: Hyperoxidized species of heme have a potent capacity to induce autoreactivity of human IgG antibodies
Authors: Marie Wiatr, Maya Hadzhieva, Maxime Lecerf, Rémi Noé, Sune Justesen, Sébastien Lacroix-Demazes, Marie-Agnès Dragon-Durey, Jordan Dimitrov *
Submitted to section: Molecular Immunology,
Major comments
(1) Dimitrov and coworkers previously reported that pretreatment of pooled IgG with heme(Fe3+) resulted in increased binding of antibodies to bacteria antigens and to intact bacteria. Further, they reported that the IgG-heme complex possesses catalytic redox-activity and acts as a potent antioxidant system. In the present study, they studied the effect of other species of heme, i.e., species that were formed after contact of heme with oxidizing agents such as hydrogen peroxide and hypochlorite anions, situations where heme’s iron could acquire higher oxidation states (hyperoxidized heme species, referred as “heme-ox” in the present study). They observed that the heme-ox has a superior capacity to heme(Fe3+) in triggering the autoreactivity of human IgG. They also demonstrated that the heme-oxo interacted with IgG at higher affinities and that this binding occurred through a different mechanism as compared to heme(Fe3+).
(2) The present results are very interesting and they would contribute to a better understanding of the interaction of antibodies with the products of pathophysiological consequence of hemolysis, i.e., hemoglobin, heme(Fe3+), “heme-oxo” species, and ROS, as the authors proposed.
(3) Accordingly, present study can be publishable in IJMS. However, before the final decision to be made, following points should be clarified.
(4) The authors assumed that the “heme-oxo” species would bind to IgG to form IgG-heme-oxo complex which is responsible for the triggering of the autoreactivity of human IgG. However, the “heme-oxo” species, which was produced in the present study by the pre-exposure of heme(Fe3+) with hydrogen peroxide or hypochlorite anions (at sub-equimolar concentration, (0.24:1), to a moderate molar excess, (25:1), concentration), would be a ferryl-heme species (either Fe4+=O, Compound I or Fe4+-OH, Compound II). Such high-valent species are known to be very unstable under ambient environments and will return to original heme(Fe3+) very quickly. Therefore, there is a possibility that even a slight excess amounts of hydrogen peroxide would be consumed up before or during the mixing experiments with IgG. How the authors confirmed that such high-valent “heme-oxo” species continue to exist during the mixing experiments with IgG?
(5) Although the authors argued that “the molecular species of heme that most efficiently induced IgG autoreactivity retained to a large extent by the integrity of the macrocyclic porphyrin structure” (page 7, lines 211-213), the evidence for this was not so strong.
(6) As indicated in Supplementary Figure 1, pre-treatment of Zn(II) protoporphyrin IX with increasing concentrations of hydrogen peroxide (0, 3.9 mM up to 1000 mM) did not cause the increase in interaction of IgG with proteins. This result suggest that the products produced by the direct attack of hydrogen peroxide to porphyrin ring are not responsible but the products produced by the catalytic activity of Fe(III) protoporphyrin IX (i.e., heme(Fe3+)) with hydrogen peroxide are responsible, as stated by the authors (page 9, lines 251-252). Therefore, considering this result and the arguments in (4), the most likely heme species responsible for the increase in interaction of IgG with proteins is a byproduct of catalytic cycle of heme and hydrogen peroxide (i.e., hydroxylated heme intermediates as seen for heme oxygenase and MhuD). I feel that this kind arguments may be necessary in Discussion section.

Author Response
(1) Dimitrov and coworkers previously reported that pretreatment of pooled IgG with heme(Fe3+) resulted in increased binding of antibodies to bacteria antigens and to intact bacteria. Further, they reported that the IgG-heme complex possesses catalytic redox-activity and acts as a potent antioxidant system. In the present study, they studied the effect of other species of heme, i.e., species that were formed after contact of heme with oxidizing agents such as hydrogen peroxide and hypochlorite anions, situations where heme’s iron could acquire higher oxidation states (hyperoxidized heme species, referred as “heme-ox” in the present study). They observed that the heme-ox has a superior capacity to heme(Fe3+) in triggering the autoreactivity of human IgG. They also demonstrated that the heme-oxo interacted with IgG at higher affinities and that this binding occurred through a different mechanism as compared to heme(Fe3+).
(2) The present results are very interesting and they would contribute to a better understanding of the interaction of antibodies with the products of pathophysiological consequence of hemolysis, i.e., hemoglobin, heme(Fe3+), “heme-oxo” species, and ROS, as the authors proposed.
We are thankful to the Reviewer for the positive assessment of our study.
(3) Accordingly, present study can be publishable in IJMS. However, before the final decision to be made, following points should be clarified.
(4) The authors assumed that the “heme-oxo” species would bind to IgG to form IgG-heme-oxo complex which is responsible for the triggering of the autoreactivity of human IgG. However, the “heme-oxo” species, which was produced in the present study by the pre-exposure of heme(Fe3+) with hydrogen peroxide or hypochlorite anions (at sub-equimolar concentration, (0.24:1), to a moderate molar excess, (25:1), concentration), would be a ferryl-heme species (either Fe4+=O, Compound I or Fe4+-OH, Compound II). Such high-valent species are known to be very unstable under ambient environments and will return to original heme(Fe3+) very quickly. Therefore, there is a possibility that even a slight excess amounts of hydrogen peroxide would be consumed up before or during the mixing experiments with IgG. How the authors confirmed that such high-valent “heme-oxo” species continue to exist during the mixing experiments with IgG?
We are grateful to the Reviewer for insightful comments about the molecular aspect of the studied phenomenon. We agree with Reviewer that hyperoxidized iron species are very transient and unstable, especially if all amounts of peroxide are catalytically consumed. Some evidence for the stability of the species formed in this work after mixing heme (Fe3+) with hydrogen peroxide came from spectroscopy experiments (Fig. 3). Indeed, measurements of the interaction of heme-ox with high affinity ligand for heme’s iron CN anions was performed at similar time-scale as the one used for preparation of heme-ox and treatment of antibodies. At these time-scale peroxide pre-exposed heme (Fe3+) was refractory to the effect of CN anions.
Another prove for role of oxidation of iron ion was obtained by the experiments where ascorbic acid was used. Indeed, the presence of ascorbate reduced the ability of peroxide pre-treated heme to induce autoreactivity of IgG.
To better clarify this point, we have now added in Discussion section a paragraph elaborating on these aspects. We highlighted there that the species formed after contact of heme (Fe3+) with oxidizing agents can be heterogenous and further studies should better characterize their nature and mechanisms.
(5) Although the authors argued that “the molecular species of heme that most efficiently induced IgG autoreactivity retained to a large extent by the integrity of the macrocyclic porphyrin structure” (page 7, lines 211-213), the evidence for this was not so strong.
We have now revised the text of the manuscript and removed this speculation. We expressed the possibility that covalent modifications of the macrocyclic structure of heme induced by the pseudo-peroxidase catalytic reaction could be alternative explanation for observed different functional activity of hyperoxidized heme species.
(6) As indicated in Supplementary Figure 1, pre-treatment of Zn(II) protoporphyrin IX with increasing concentrations of hydrogen peroxide (0, 3.9 mM up to 1000 mM) did not cause the increase in interaction of IgG with proteins. This result suggest that the products produced by the direct attack of hydrogen peroxide to porphyrin ring are not responsible but the products produced by the catalytic activity of Fe(III) protoporphyrin IX (i.e., heme(Fe3+)) with hydrogen peroxide are responsible, as stated by the authors (page 9, lines 251-252). Therefore, considering this result and the arguments in (4), the most likely heme species responsible for the increase in interaction of IgG with proteins is a byproduct of catalytic cycle of heme and hydrogen peroxide (i.e., hydroxylated heme intermediates as seen for heme oxygenase and MhuD). I feel that this kind arguments may be necessary in Discussion section.
We fully agree with the Reviewer that the covalent modification of heme structures that depend and are most likely formed a as a result of pseudo-peroxidase reaction (as the case of Tyr radicals in hemoglobin) might be generated in our experimental system. To better explain these aspects in the manuscript, we have now extended the Discussion section by adding a paragraph discussing the effect of peroxide on heme.

Reviewer 2 Report
In the present study, the Wiatr et.al. are trying to elucidate the Autoreactivity of human IgG triggered by oxidized species of heme which are created in pathological conditions. There are following points which needs to be considered:
Overall, authors represented the data in the raw form without any statistical analysis. The lack of p values makes it hard to infer the results.
The additional in vitro studies with cell lines and in vivo studies with pathological mouse model is needed.
In Figure 1) The significant P values are not shown on any of the graphs. The statistical significance of comparison is needed to determine the difference between the binding of both the heme and heme-ox. (e) Immunoblot analysis needs to be replaced by another blot with more convincing results. (f) Microscopic scale is missing in the figure,
In Figure 2) Statistical p values are missing in all the graphs, in Figure 2 A, its difficult to distinguish the difference between native IgG and treated IgG, kindly represent it in alternative way to make the difference evident.
In Figure, 3a the difference between the two graphs has not been stated, please add the difference between two graphs, also the graph axis labels are missing. In figure C, multiple comparisons could also be added among the groups.
Under the section 2.4, the rationale behind choosing the prototype IgG1 needs to be added and other IgG sub types such as IgG2, IgG3 and IgG4 could have been also added in the analyses. As the comparisons are done between heme and ox-heme the scales of the depicted graphs should be same and the raw values can also be depicted in the graphical format where the statistical comparisons can be done among the groups.
Currently, the data presented in all the figure is in very raw state and needs an additional processing to be subjected to statistical analysis, without the p values it becomes difficult to back the conclusions stated by the authors.
The in vivo experiments are missing in this study, it would be more convincing if authors could replicate the results in pathological mouse model.
In Discussion, the significance of porphyrin ring structure needs to be added. Also, discussion about the impact of porphyrin ring disintegration.
Author Response
In the present study, the Wiatr et.al. are trying to elucidate the Autoreactivity of human IgG triggered by oxidized species of heme which are created in pathological conditions. There are following points which needs to be considered:
Overall, authors represented the data in the raw form without any statistical analysis. The lack of p values makes it hard to infer the results.
We agree that the statistical analyses of the data will make the manuscript more reliable. We have now performed statistical analyses of our data and presented p values on the respective graphs. The statistical tests used were cited in the legends to figures. The statistical analyses confirmed that the effects of heme-ox on the reactivity of human IgG are significantly different from those of heme (Fe3+).
The additional in vitro studies with cell lines and in vivo studies with pathological mouse model is needed.
We fully agree that further studies with cell lines and animal models would be of large interest. However, we consider these studies out of the scope of this work as it is mainly dedicated on description of the phenomenon and analyses of its biochemical mechanisms. It is important to note that in a previous study we have analyses the effect of hyperoxidized heme species in cellular model (See PMID: 31078967). In this study we demonstrated that the exposure of human endothelial cells to heme-ox results in a potent activation of complement system. Higher recruitment of complement proteins on the surface of endothelial cells might have important pathological repercussions leading to cellular damage. Importantly, this study had also shown that human therapeutic pooled immunoglobulins (IVIG) could directly scavenge heme-ox and thus protect endothelial cells from the attack by complement system.
To acknowledge this Reviewer’s remark, we have now referred in the revised version of the manuscript to the study using a cellular model. In addition, we stated as perspective that the physiological significance of phenomenon should be elucidated by appropriate in vivo models.
In Figure 1) The significant P values are not shown on any of the graphs. The statistical significance of comparison is needed to determine the difference between the binding of both the heme and heme-ox. (e) Immunoblot analysis needs to be replaced by another blot with more convincing results. (f) Microscopic scale is missing in the figure,
We have now performed statistical analyses and added p values to the figures. The figure of immunoblot was edited (enlarged and contrast modified) to clearly discern the difference in the reactivity of heme (Fe3+) and heme-ox – treated polyclonal human IgG to the numerous proteins present in the endothelial cells lysate. On the pictures from immunoforescence analyses with Hep2 cells and human neutrophils we indicated the used magnification of the microscope.
In Figure 2) Statistical p values are missing in all the graphs, in Figure 2 A, its difficult to distinguish the difference between native IgG and treated IgG, kindly represent it in alternative way to make the difference evident.
We have added p values from statistical analyses concerning the Figure 2b.
Regarding Figure 2A, the elution profiles of native and heme (Fe3+) or heme-ox – treated IgG are overlapping. This result clearly indicates that there is no difference in the molecular composition in IgG preparation after exposure to heme species. As there is no noticeable difference in the elution profiles, we are not able to avoid overlap of the curves and present them separately.
In Figure, 3a the difference between the two graphs has not been stated, please add the difference between two graphs, also the graph axis labels are missing. In figure C, multiple comparisons could also be added among the groups.
We have now clearly stated in text the difference between the two graphs in Figure 3a. Moreover, we added X and Y titles on the figure. Recommended statistical analyses were performed and p values added on the figure.
Under the section 2.4, the rationale behind choosing the prototype IgG1 needs to be added and other IgG sub types such as IgG2, IgG3 and IgG4 could have been also added in the analyses. As the comparisons are done between heme and ox-heme the scales of the depicted graphs should be same and the raw values can also be depicted in the graphical format where the statistical comparisons can be done among the groups.
The rationale for use of IgG1 as a model is two-fold. Firstly, this is the most abounded subclass of human IgG (about 70 % of total IgG). Secondly this subclass is the preferred subclass for therapeutic antibodies, owing to its high stability, homogeneity and biological activities. Importantly most of our experiments are performed with pooled human IgG which represents natural distribution of IgG1, IgG2, IgG3 and IgG4 in human immune repertoires.
The spectroscopy data in section 2.4 there are difference in the spectral profiles, besides the differences in the intensity. The fine spectral characteristics are very informative for the molecular microenvitroment of heme. The scales on Fig. 4 for pooled IgG and monoclonal IgG are different because we did not directly compare them and wanted to highlight the spectrum and differences between heme(Fe3+) and heme-ox within each type of immunoglobulin.
Currently, the data presented in all the figure is in very raw state and needs an additional processing to be subjected to statistical analysis, without the p values it becomes difficult to back the conclusions stated by the authors.
We have now performed statistical analyses and added respective p values.
The in vivo experiments are missing in this study, it would be more convincing if authors could replicate the results in pathological mouse model.
We fully agree that further studies with animal models would be of large interest for understanding the pathophysiological relevance of the effects of heme-ox on immunoglobulins. However, we consider these studies out of the scope of the present work as it is mainly dedicated on description of the phenomenon and analyses of its biochemical mechanisms.
To acknowledge the Reviewer remark, we have now stated as perspective that the physiological significance of phenomenon should be elucidated by appropriate in vivo models.
In Discussion, the significance of porphyrin ring structure needs to be added. Also, discussion about the impact of porphyrin ring disintegration.
We have now revised the Discussion section of the manuscript and presented additional information about the potential presence of covalent modification of porphyrin macrocyclic structure in heme-ox. The potential roles of these modifications and those of the high valent oxidation states for the induction of novel antigen-binding specificities of antibodies was also discussed.

Round 2
Reviewer 2 Report
Authors have added the statistical analysis in the revised manuscript, the pairwise comparisons among the groups were not shown in all the figures. The description of the significant values should be added in results section. I would suggest authors to refer other publications and follow the in general representation of state of the art statistical analysis.
Under the materials and methods section authors should also add the separate section for statistical analysis. What all test were used and what software was used to perform the tests should be added.
The blots are still not convincing. The original blot should be labeled properly.
Author Response
Authors have added the statistical analysis in the revised manuscript, the pairwise comparisons among the groups were not shown in all the figures. The description of the significant values should be added in results section. I would suggest authors to refer other publications and follow the in general representation of state of the art statistical analysis. Under the materials and methods section authors should also add the separate section for statistical analysis. What all test were used and what software was used to perform the tests should be added.
To address the issues pointed by the Reviewer, we have now added result from statistical analyses on Fig. 2d (which lacks significance between heme and heme-ox treated groups). We also added description of significant values in the Results section of the manuscript. A new subsection describing applied statistical analyses and the software used in the manuscript was added to the Materials and Methods section.
The blots are still not convincing. The original blot should be labeled properly.
As recommended by the Reviewer the image of immunoblot has been now edited to clearly label all condition. We displayed the concentrations of polyclonal IgG directly in the figure.